# “Es Muy Tranquilo Aquí”: Perceptions of Safety and Calm among Binationally Mobile Mexican Immigrants in a Rural Border Community

**DOI:** 10.3390/ijerph19148399

**Published:** 2022-07-09

**Authors:** Rebecca M. Crocker, Karina Duenas, Luis Vázquez, Maia Ingram, Felina M. Cordova-Marks, Emma Torres, Scott Carvajal

**Affiliations:** 1Department of Health Promotion Sciences, Zuckerman College of Public Health, University of Arizona, Tucson, AZ 85724, USA; krduenas@arizona.edu (K.D.); felina@arizona.edu (F.M.C.-M.); 2Campesinos Sin Fronteras, Somerton, AZ 85350, USA; lvazquez@campesinossinfronteras.org (L.V.); etorres@campesinossinfronteras.org (E.T.); 3Arizona Prevention Research Center, Department of Health Promotion Sciences, Zuckerman College of Public Health, University of Arizona, Tucson, AZ 85724, USA; maiai@arizona.edu (M.I.); carvajal@arizona.edu (S.C.)

**Keywords:** Mexican immigrants, U.S.-Mexico borderlands, community perceptions, safety, violence, well-being, binational, external referents

## Abstract

Perceptions of community can play an important role in determining health and well-being. We know little, however, about residents’ perceptions of community safety in the Southwestern borderlands, an area frequently portrayed as plagued by disorder. The qualitative aim of this community-based participatory research study was to explore the perceptions of Mexican-origin border residents about their communities in southern Yuma County, Arizona. Our team of University of Arizona researchers and staff from Campesinos Sin Fronteras, a grassroots farmworker support agency in Yuma County, Arizona, developed a bilingual interview guide and recruited participants through radio adds, flyers, and cold calls among existing agency clientele. Thirty individual interviews with participants of Mexican origin who live in and/or work in rural Yuma County were conducted remotely in 2021. Participants overwhelmingly perceived their communities as both calm and safe. While some participants mentioned safety concerns, the vast majority described high levels of personal security and credited both neighbors and police for ensuring local safety. These perceptions were stated in direct contrast to those across the border, where participants had positive familial and cultural ties but negative perceptions regarding widespread violence. In conclusion, we argue that to understand environmental factors affecting health and well-being in Mexican immigrant populations, it is critical to examine the role of binational external referents that color community perceptions.

## 1. Introduction

The extent to which individuals hold positive perceptions of their neighborhoods and communities may have important implications for their health and wellbeing [1]. Growing awareness of the importance of upstream influences has led to increased interest in “place effects”, or the impact of our social and physical environments on human health [2]. Such ties have been explored in the literature about geographies of health and well-being, where the material and socioeconomic conditions of one’s lived environment have been assigned a seat at the table next to individual and family factors as determinants of variation and inequities in individual wellbeing [3,4]. While much of this research has addressed negative contributing factors such as violence, lack of infrastructure, and unsightly surroundings, the theory of “therapeutic landscapes” has been widely employed to identify the places or milieus (both physical and psychological) that can, by contrast, promote healing and wellness [5].

Perceptions of a community may be influenced by a broad variety of factors attendant to individual and community level priorities and needs for wellbeing [6]. These include aesthetic and safety features such as lighting and sidewalks, housing style and affordability [7] and the availability of local resources like parks and schools [8]. It also incorporates less tangible socio-cultural aspects founded in community history, ethnic identity and social networks, with social cohesion acting as a major contributor to wellbeing [9]. In addition, perceived personal safety has been associated with enhanced wellbeing and health promoting physical activity [10], whereas fear and experiences of crime have a negative influence on how individuals assess their neighborhoods and can carry long-term health consequences [11,12].

While the U.S.–Mexico border region has been a locus of media attention characterizing the region as unfavorably affected by uncontrolled migration, “spillover” drug related violence from Mexico, and poor public safety, we know little about how border inhabitants themselves perceive and experience their lived environments. This scarcity of data on border residents limits our access to what Castañeda and Chiappetta (2020) term an “authentic interpretation of border security,” independent from sensationalized media portrayals [13] (p. 11). Crime statistics, however, indicate that the region has been mischaracterized. They demonstrate that cities and settlements on the U.S. side of the international border have lower than average crime rates as compared to other regions of the U.S. [14,15]. Moreover, crime rates in U.S. border cities are decreasing faster than in the rest of the country, even at a time when drug war related violence has spiked in many Mexican towns and cities right across the border [16,17].

Moreover, the limited available data indicates that most border residents have positive perceptions of their communities and local safety, reflecting these official low crime rates [13]. Communities in the U.S. borderlands are populated largely by Mexican origin residents, and their shared sociocultural identity alone may confer a health advantage [18]. Evidence from El Paso, Texas indicates that the primarily Mexican origin residents positively described their city as “safe”, “calm”, and “good” and did not perceive that community safety was dependent on the border wall [13,19]. This is somewhat surprising given that Mexican immigrants are often disproportionately vulnerable to attacks and robberies and are less likely to report crimes to the police due to concerns over immigration status [20,21].

It has been suggested that when interpreting place-based perceptions of safety and other community attributes, it may be important to consider changes in referents due to residential mobility [3]. This is particularly relevant in studies along the U.S.–Mexico border, where a significant portion of residents were born in Mexico and have regular and constant cross-border exposures. A binational lens has been incorporated elsewhere to better understand immigrants’ perceptions of other critical health-related issues, including blocked access to medical care post-migration, the relevance of preventive dental care, and ties between trauma and mental health [22,23,24]. Moreover, for Mexicans living in the United States, the change in place of residence often entails a significant loss of civic and human rights due to the dearth of legalized immigration pathways for them. The realities of living as long-term “undocumented” residents of a foreign country experiencing a wave of anti-immigrant public sentiment and exclusionary legislation likely color all nature of community perceptions.

Further exploration of how border residents perceive their communities within this binational context is warranted, particularly given the link between perceptions of social and physical environments and human health. In this article, we present the qualitative results of a study that aimed to explore the perceptions of Mexican-origin border residents about their communities in southern Yuma County, Arizona.

## 2. Materials and Methods

### 2.1. Study Aims

This article presents data from the first data collection phase of a community-based participatory research (CBPR) project, taking an ecological approach to identifying sources of stress and resilience among Mexican-origin persons in Yuma County, Arizona [25,26].

### 2.2. Study Design

This multi-year, mixed-methods study employed a CBPR approach bringing together researchers from the University of Arizona (UA) and a workforce of community health workers (known in Spanish as *promotoras*) and behavioral health technicians from Campesinos Sin Fronteras (CSF). CSF is a grassroots community organization in Yuma County, Arizona founded by farmworkers to address unmet health and social needs they identified in their community. The 6th author is CSF’s current Executive Director and one of its founders. The study’s CBPR approach was based on over 20 years of partnership addressing the health and safety of agricultural workers in the region. This study’s focus on stress and resilience among Latino/a adults grew out of the team’s previous work on farmworker experiences of stress and discrimination, and ongoing concern from CSF regarding poor access to mental health services and support [27]. In addition to increasing the quality of the research, this strong academic–community collaboration ensures that the research is responsive to community priorities and provides community benefit [28,29].

The development of aim 1 of this study began in August 2020 with weekly meetings between the UA research team and the CSF team. These meetings were guided by reviewing the study’s stated aims but were held in an open format that encouraged all members of the study team to discuss key areas of interest relating to sources of stress and potential areas of resilience in southern Yuma County. These meetings were held remotely via zoom and participants were encouraged to speak in their preferred language and a bilingual member of the team provided translation in the Zoom chat feature. This design was aimed to ensure that all team members felt comfortable expressing their ideas, opinions and expertise to the entire group. Team building and team collaboration was also enhanced by an immersion tour of Somerton and San Luis, Arizona hosted by CSF in September 2021, aimed to build cultural and community specific familiarity.

Based on a careful review of notes summarizing the priority areas and themes identified during team discussions, the lead author developed a draft qualitative interview guide in English and Spanish. This draft was reviewed by the entire team for representation of salient themes and cultural and community congruence, and the Spanish version was edited by native speakers for regional appropriateness. The resulting semi-structured interview guide (see Appendix A) focused on five domain areas in the context of living on the border and in a binational environment: 1. health concepts, 2. personal health and wellbeing, 3. sources of resilience, 4. sources of stress, and 5. community. We also included a short demographic questionnaire aimed to capture individual characteristics including gender, birth year, marital status, household members, family size, binational connection, and employment.

### 2.3. Sample Participants

Participants for the study (N = 30) were recruited by the CSF promotora workforce through existing CSF client networks, phone calls, flyers placed in the CSF waiting room, radio advertisements on the CSF radio station, a feature run in a local Spanish newspaper, and word of mouth. Inclusion criteria included that the person: (1) be of Latino/a origin, (2) live in the border region of either southern Yuma County, Arizona, or northern Sonora, Mexico, (3) be between the ages of 18–70, and (4) have access to a telephone or computer for Zoom-based calls. After describing the study to potential participants, the promotoras screened them for eligibility. Those who were interested and eligible were then consented verbally, provided with a copy of the consent disclosure for their records, and scheduled for an interview as soon as possible.

### 2.4. Data Collection

While data collection was originally planned to be in person, the team switched to password protected Zoom audio calls due to the onset of the COVID-19 pandemic. In order to assist participants with Zoom technology, CSF promotoras sent interview reminders, provided instructions on connecting to the Zoom platform, and completed a follow-up call after the interview was completed. Three members of the UA team and two CSF behavioral health technicians led the qualitative interviews. All five were bilingual and experienced at conducting culturally congruent interviews with Latino/a populations. In preparation for data gathering, the two behavioral health technicians received training on protection of human subjects, ethics, and providing informed consent. In addition, the lead author conducted a bilingual interview training for the interview team.

The promotoras piloted the semi-structured interview questions with Spanish speaking CSF staff. Once participant recruitment began, we implemented the interview guide with three participants and then reviewed the interview guide implementation to ensure fluidity and the tailoring of questions to cover all areas of study aims. Interviews were conducted in the participants’ preferred language (29 in Spanish and one in English) and lasted between 40–80 min. Participants were given a $40 Walmart gift card incentive for their time.

### 2.5. Data Analysis

Audio files of the interviews were transcribed in the language spoken by a bilingual, bicultural graduate student and a professional transcription company (GMR Transcription). Interviewers then conducted quality assurance checks of transcriptions. After reviewing transcripts, the research team agreed that data saturation had been reached and salient themes were evident. At this point, transcripts were uploaded to Dedoose data management software (www.dedoose.com, accessed on 19 April 2022) and two UA members of the research team independently coded 15 interview transcripts each (see Appendix A). Notations of the rationale for additional codes and revised codes were discussed for consensus, and following completion of coding, the research team discussed the most prominent findings, surprises in the data, and weighed the relative impact of the findings to prioritize dissemination in the framework of the team’s research priorities and CSF’s areas of interest.

### 2.6. Ethical Considerations

This study’s CBPR approach facilitated our recruitment process and ensured open and culturally consonant communication with research participants. Given the sensitive nature of questions surrounding immigration, we encouraged participants to verbalize any discomfort throughout the interview process and assured them that they could refrain from answering any question or stop the interview at any time. At the time of receiving their incentive, the promotoras inquired with each participant about their experience in the interview and accepted any feedback that was provided. This study was approved by the UA human subjects’ protections review board (2003462440). To protect participant privacy, pseudonyms are used throughout this article.

## 3. Results

This study sample of Mexican origin participants included 17 women and 13 men, with an average age of 35. Half of participants were married, while the remaining 15 were primarily single or in domestic partnerships, and most (23/30) had at least one child. Most participants (23/30) were born in the Mexican border states of Sonora or Baja California Norte, three were from other states in Mexico (Guanajuato, Michoacán and Sinaloa), and four were born in the U.S. border states of Arizona and California. Most participants lived in Yuma County, Arizona (17 in San Luis, Arizona, six in Somerton, and three in Yuma), while four lived in the Mexican border city of San Luis Río Colorado, Sonora. Occupations in the sample varied and included retail positions, maintenance, and clerical work, but the most common were in agriculture (8/30) and teacher or teacher’s aide (5/30). Five participants were either unemployed or stay-at-home mothers. For a full presentation of sample demographic characteristics, see Table 1.

Study participants overwhelmingly reported that the small towns of Somerton and San Luis, Arizona provided safe and calm environments in which to live and raise families. They frequently described their communities as “tranquilo” (meaning calm, tranquil, peaceful, or quiet) and safety and calm were identified as major advantages to living in southern Yuma County (For original quotes in Spanish, see Appendix A). Three primary themes emerged in relationship to these findings: (1) Perception of Calm, with a sub-theme of how calm environments reduce stress; (2) Comparing Safety in Yuma County and Mexico, with sub-themes on safety concerns in Yuma County, comparisons to higher levels of violence in Mexico, and the roles of neighbors and police in promoting safety; and (3) Appreciation of Safety in Southern Yuma County, with a sub-theme on the compromises made for gaining safety.

### 3.1. Perceptions of Calm

Most participants living on the U.S. side of the border described the places where they lived as calm and peaceful, perceptions that were often related to these places being relatively small towns in a predominantly rural region. This notion of “small town” was tied to several specific benefits, including the absence of excessive noise, crowded environments, and dangerous or frustrating transportation issues. Luis, a resident of San Luis, Arizona expressed: “I think that, first and foremost, what I like is that it’s a quiet city, right? There isn’t—it’s not a very big city, it’s not noisy.”

The region’s sense of calm was often compared positively to other larger cities in both Mexico and the US. Most participants had had prior experience living in larger cities, most commonly San Luis Rio Colorado, Sonora, the fourth largest city in state of Sonora, and the large urban center of Mexicali, Baja California Norte. Others had been exposed to other large cities in Mexico or had traveled to work in urban areas in the U.S. southwest and west, such as Los Angeles or Salinas, California, and Tucson or Phoenix, Arizona. Manuela explained that while San Luis, Arizona was larger than Somerton, “It is quieter than, I don’t know, I have been to California and places like that, Phoenix, and it seems to me like there is a lot of traffic. Here I can go out in the street without being scared of, well of driving, because I’m a really nervous driver.” While some acknowledged that the size of their towns limited opportunities for entertainment and recreation, many participants felt there were enough stores, restaurants, and parks to serve their basic needs.

In addition to appreciating the small size of the communities where they lived, participants also attributed the region’s calm environment to the fact that their neighbors were either older or lived in family units, were work-focused (“gente de trabajo”), and generally kept to themselves. Anita, a married woman living in Somerton said: “In terms of tranquility, here where I am living, I live around many elderly people, so it is a very quiet community. I used to live in Yuma and it was like a trailer park where I lived, and there was a lot of disorder, a lot of problems. Not here, here everything is really quiet.” Many participants specifically acknowledged the lack of altercations with their neighbors and the fact that no one appeared interested in starting problems. For example, Francisca, a widow who had lived in the region for over 30 years, commented that: “Well it’s very peaceful, very calm, because my neighbors are not rowdy. I have never had a fight with anyone there. Very peaceful, very calm.”

#### A Calm Environment Reduces Stress

Participants frequently commented that the small and calm nature of their communities created ease in their daily lives and made it less stressful than living in a large city or other places where one had more daily concerns. Many mentioned specific practical advantages of living in a quiet place that reduced stress, including ready access to stores, restaurants and other facilities. Adán, who moved to Somerton from Yuma 22 years ago stated: “But what can I tell you? The market is just a three-minute walk away, I have stores here. I live right along the main street and we have everything close by. You don’t get stressed out about the traffic. Here when you go to the store, you walk.”

Other participants commented more generally that living in a small town afforded a slower pace of life. When asked if the stressors in her community were similar to those experienced elsewhere, Somerton resident Monica replied: “No I don’t think they are the same because I feel that both the community and the city are a bit calmer and there are other places where everything is faster paced. So, I don’t think they are the same.” Some participants noted that this slower pace both reduced stress and enabled them to enjoy their daily lives more fully. Evaristo, who had lived in the United States for 20 years commented that in general life north of the border felt very rushed and focused around work, but that in San Luis: “Here I feel that, yes, you enjoy it a little bit more. But I don’t know exactly how to explain it but I feel calmer in this area than in a city.”

For Somerton resident Olger, a married father of three originally from the Sonoran capital of Hermosillo, this calm translated as his “comfort zone.” He explained:

“It’s not very stressful here, because it is basically a small town. I would say this place is not very stressful … Sometimes we have talked about buying a house and we ask ourselves: ‘where? Where would we go?’ And I have always said: ‘well not Yuma, not Yuma.’ Because it could be really pretty in some parts, in some parts quiet. But you do feel the stress of living in the city a bit more because of all the traffic, perhaps lots of malicious people, etc. And here it is really a little calmer. Yes, it’s basically, it’s like my space, my comfort zone, and it’s even like—I live in apartments, I mean, I don’t have a house still, right? And in fact, I live where I work. And I feel comfortable and safe here. Why? Because I am familiar with my surroundings, I know all the people here, the neighbors, I know how the facilities are. So, I don’t have much to worry about.”

For most participants, limiting stress was understood to be an important component to mental and physical health. For example, Manuela, an agricultural field-worker who lived in Yuma, explained: “I think that living a life with lots of stress deteriorates you, it wears you down.” Because of this, many participants believed that finding calm was a critical element of health maintenance.

### 3.2. Comparing Safety in Southern Yuma County and Mexico

#### 3.2.1. Safety Concerns in Southern Yuma County

A few participants raised safety concerns about their communities in Yuma County, some of which were common to the broader region, such as car accidents and gang activity, while others were closely tied to the region’s proximity of the US–Mexico border, such as recruitment of youth into the drug trade, undocumented migration, and border surveillance. For example, Rosie, whose home was very close to the border wall and had an unfenced yard, described:

“Immigration officials go there, behind my house. Helicopters are out starting early in the morning … and you can’t know if there might be, if someone might be running or have a gun. That has never happened because I don’t think they come to do those things. They cross because they want to get ahead. But still, there have been many cases in which they don’t come armed but the police themselves can shoot off their guns just because they want to catch someone.”

Others mentioned concerns about “spillover violence” from Mexico, noting the disappearance and subsequent murders of two young women from the area in 2021, a crime that participants felt was reminiscent of violence in Mexico. Chucho, a young man originally from Los Angeles, said that he was starting to hear about “the violence, like cartels doing like killings and gun shootings across the border … like that type of like news has been increasing and that like the only worry is like, when will it spill over to this side?”

#### 3.2.2. Comparison to Higher Violence Levels in Mexico

Participants named far more safety concerns, however, when speaking about Mexico and particularly the northern border region. Most participants (23/30) were originally from the border states of Sonora and Baja California Norte, and had witnessed dramatic increases in drug-related violence over the past decades. While the majority of participants (26/30) now lived on the US side of the border, they had frequent exposure to Mexico through regular border crossings to visit with family, friends, and significant others, attend celebrations and family events, doctors’ appointments, and shopping. Those who lived in San Luis, Arizona described living “right on the border”, and being able to see Mexico and the busy crossing lines from their homes. The four participants who lived in Mexico crossed the border daily or weekly.

These binational cultural and community ties afforded participants first-hand experiences with issues around compromised public safety in Mexico. For example, Ana, originally from San Luis Río Colorado, explained: “Right now there is a lot of crime, and they say this is because everyone wants to take control of the borders. Right now in San Luis Río Colorado there has been a lot of, well something that we had never seen here: murders, kidnappings in broad daylight.”

Many participants described a declining arc of public safety in Mexico due to the drug trade that made living there and even visiting potentially risky. They mentioned threats to security including vandalism, robbery, violent attacks, gun fights, accidents, killings, and kidnappings. Diego, who moved to San Luis four years ago from Mexicali, explained: “Oh yes, it’s really different by comparison to where I lived in Mexico, because there it was, well where I lived it was a really dangerous block. There were many people doing drugs … and just leaving my house you could get assaulted. That happened many times, not to us, but to other people.”

#### 3.2.3. Perceptions of Relative Safety in Southern Yuma County

Participants’ binational frame of reference had a direct impact on the way they perceived security in the cities and towns on the U.S. side of the border, with the vast majority reporting that the communities in southern Yuma County felt very safe by comparison. Monica, originally from Mexicali, Baja California Norte, said: “Look, the community where we are now is actually calm. But if you’re talking about Mexico, Mexico is really hectic. In Mexicali everything is go, go, go. But everything in Yuma County is peaceful. Here there are more older people so it’s quieter, all things considered.” Often participants living in the U.S. could not think of anything that made them feel unsafe in their communities and described a deep level of personal security both in terms of protection from acts of violence as well as the security of their possessions.

Elia, lamented that the Sonoran ejido of her childhood had been a safe place but that public safety had declined dramatically since then. She stated: “Here in San Luis, Arizona, I feel very safe. There’s not, how can I say, like in Mexico, so many gangs, so many addictions, so much alcohol, so many bad people in the street. I feel very safe, very safe here, more peaceful. I can, yes, I can leave my girls outside while I go in for a drink of water or something. I have my fence and everything, but I can leave them outside briefly.”

Participants’ conceptualization of regional safety was often defined by the absence of major violent events such as shoot-outs, kidnappings, and murders. Angeles, a young woman originally from the city of Mexicali, Baja California Norte, explained that what she liked about San Luis, Arizona was that: “You almost never hear about there being problems. Lately there are a lot of car accidents, but I mean, you don’t hear about gunshots, horrible things like that, they’re unheard of.”

Francisca indicated that the lack of such violent outbursts gave her a sense of freedom with her children. She explained: “Here, one can comfortably take the kids to the park or get them out on a walk. Yes, it’s very rare that you hear about anything bad here in San Luis, Arizona, like that someone was killed or run over or something. There [in Mexico] no. There no. There you look at someone the wrong way and you don’t make it.”

#### 3.2.4. The Role of Neighbors in Promoting Safety

In reflecting on why their communities in Yuma County felt safe to them, participants frequently commented on the helpful roles played by their neighbors. While many participants reported not knowing their neighbors well, most described generally positive relationships and appreciation of the fact that their neighbors did not start verbal or physical altercations with each other. Tomás explained: “the families all keep to themselves.”

Moreover, participants remarked that their neighbors supported community safety by taking good care of their own properties, alerting each other to any issues, and watching out for each other’s personal safety and belongings. For example, Sabina noted that her neighbors were always willing to lend a hand. She explained: “I truly feel that this is a safe community. I feel so safe here where I live, so safe that—and this is my fault—that sometimes I say: ‘I left the bicycle outside without putting the outside light on. And the bicycle is still there in the morning!’ (laughter) … so I feel that, ay, yes, my neighbors will watch it for me.”

A few participants expressed that the fact that most local residents were of Mexican origin and Spanish speaking greatly eased their ability to communicate and support each other in keeping the area safe. For example, one thing that Tomás appreciated about his home in San Luis, Arizona was the constant influx of new immigrants from Mexico. He explained: “What makes it different is that new young people from Mexico are always arriving. So, there is a little more Mexican culture here. And they settle here, and you could say that that is why we make a good match, because we basically have the same roots and the same behaviors.”

Diego, who had moved to San Luis four years prior described a neighborhood Facebook group that added an additional layer of cooperation around basic safety and care-taking. He said that through that group: “We also make sure that, like if they see something strange, then they let us know on there, or if a pet got loose and they find it somewhere else, you know, they also notify us, things like that. We are, like, in close communication.”

#### 3.2.5. The Role of Police in Promoting Safety

Participants also noted that local police forces played a critical role in making them feel safe and protected. Given that our team had previously documented stress deriving from the multiple overlapping jurisdictions of police and border authorities in Yuma County [30], our study team included the following question to directly assess the presence of and impact of such stress: “Do you feel that the presence of border patrol and police is a cause of stress for you or members of your family or community?”

Contrary to what we expected based on this prior research, the majority of participants replied that the police played a positive role in regional security. Francisca, a woman in her mid-40s who lived in San Luis responded: “Well, not really. I think they are a means of security for the community and for those around us on both sides of the border.” Malena, who was originally from central Mexico, felt similarly. She said: “The police control makes me feel safer. Yes, I think that, as far as I can see, they do their job well. And, what can I say? Well, I don’t know, I am very at ease here.”

Several participants expressed appreciation for the constant presence and proximity of the police forces in the region, with a few mentioning being glad to have members of the police force as their close neighbors. Ivis, a mother of three who lived in Somerton stated: “Yes, it is calm here, I feel safe. And then I also have neighbors who are police here. I have neighbors who are police, I have five neighbors who are police.” Another Somerton resident, Adán, remarked that: “It makes me feel safer here because, just 30 seconds away, we have the police and fire department. That makes me feel very safe, because it’s not as if just because we live here nothing more will ever happen, right? But that gives us a lot of peace of mind.”

Others commented on how regularly the police passed through the areas where they lived. Pancho, originally from Culiacán, Sinaloa expressed: “It makes me feel safe that here there is always, always really constant surveillance. Even though I live on Main Street, on the main drag, every day at the same time a patrol car passes by … always doing their rounds.”

Several participants remarked specifically on appreciating the responsiveness of the police forces, an observation that was linked both to the towns being small and to the professional training of the local forces. Sabina, a preschool teacher who had lived in San Luis for 20 years, said: “I feel protected in this community. I think the police department does its job well. I think that when people need to call them, they are there to assist them, because it’s not a very big community.” Ernesto, a young man born in U.S. but with significant cross-border family and cultural ties stated: “I feel like it makes me feel safe that here in San Luis, Arizona you can call 911 for an emergency and they respond very quickly. The officials here are, yes, they are very well-equipped, very well-trained professionals.”

While most participants referred to the local police agencies of their towns in their responses, a few mentioned also welcoming the presence of other police jurisdictions, including those from the nearby Cocopah Indian Tribe as well as federal forces. For example, Luis, a young man employed in the lettuce fields, explained how more soldiers and National Guards had recently been sent to the border near his home in San Luis, Arizona. He said: “They say there is more mobilization around public safety, right? Or of the armed forces. And the truth is that you start to feel a little more at ease, because, I mean, like when they started to arrive, the level of insecurity declined, isn’t that right?”

### 3.3. Appreciation of Safety in Southern Yuma County

When asked what they liked about where they lived, participants living in southern Yuma County overwhelmingly expressed great appreciation for how safe they felt in their small communities. Many participants offered specific examples of how they had reduced their levels of precaution and vigilance since moving to the area due to their perceived low risk of personal attacks and robbery. For example, Ami a married mother of three children stated: “I feel—let me give you an example—I feel free to have my car windows down without having to look all around me. I feel calm here, at ease.”

Participants recognized that the lack of violence, vandalism, and traffic danger positively affected many areas of their daily lives, allowing them to feel at ease while going outside to walk, shop, and eat, and being on the street. For example, Tomás stated that: “In terms of tranquility, before the pandemic one could go to the parks without any problem at all. There is no uncontrolled vandalism.” Likewise, Mario, a married father living in San Luis, commented that the lack of traffic dangers facilitated walking in public places. He said: “The good thing is that [San Luis] has, that one can walk at ease in the streets. The cars don’t go fast, they respect the traffic signals, they respect you. It’s very good.”

This sense of calm also extended to participants’ homes and personal belongings. They described being surprised upon realizing that their belongings were safe when left unsecured, and enjoying the ease of not always having to lock homes and cars. Evaristo, a fieldworker with two children stated: “It’s simply the fact that I have felt so comfortable when I go out. I come home and sometimes I say to myself ‘ay, the door was left open, and ay look, what happened?’ And how peaceful it feels to go out and know that you can even not lock up, because you know that everything will be okay.”

Some also described novel feelings around their own personal safety in their homes, including the freedom to come home alone and unaccompanied, even at night. Sabina, a single mother of five children explained: “I feel safe, so safe that sometimes I have to go to a training or something, and sometimes I forget to turn on the light. And I don’t feel scared coming home, I don’t come home like ‘oh no, my house is dark. Let me call someone to see if they can accompany me home.’ No, with all the trust in the world, I open the door to my house and I turn on the light.”

#### Making Compromises for Safety

In several instances, participants indicated that their appreciation of the region’s safety and calm had influenced their decision to move to or remain in the area, despite noting its disadvantages in other areas. Some participants indicated, for example, that they were willing to forego better economic opportunity and more developed infrastructure and services potentially available in other areas of the United States in exchange for a safe local environment. Belem noted: “Here the community is very peaceful, where we live you rarely hear much about, like in other places that they stole something or there are lots of gangs, almost nothing like that. It’s like, even though the situation here in terms of work and pay is not very good, you feel like it’s a good place to raise your kids, all considering.”

Several participants recognized that moving to the quiet communities of southern Yuma County led to a loss of social cohesion or “convivencia” (shared living experiences) they had enjoyed in Mexico. For example, Linda explained that to her what was missing from her community was: “a little more life, more of that Mexican part—or maybe another kind of culture— but it’s almost, almost like you come here to Somerton and, it’s as if, like everyone is shut away, everyone is closed off. Even before COVID, everyone was shut away in their houses.” Likewise, Vanessa noted of her home in Somerton that: “The other place I lived before was like a million times better. But the truth is, the environment around there was a little more [dangerous]. And so, I think that is the most important factor that makes me—even though I know that it’s like better there—that I still prefer to be here.”

Sabina concluded that the calm and safe environment of San Luis, Arizona was a deciding factor for her family. She stated: “It’s a quiet town. If one wants to live peacefully, I think it is a peaceful town. I compare it to San Luis Río Colorado, and I can say that for the calm and safety, I’ll stay in Arizona.”

## 4. Discussion

Over the past decades, there has been growing recognition that place and people’s perceptions of it are woven into the fabric of what creates health-inducing environments. Moreover, individuals’ experiences of the communities and neighborhoods in which they live are contributing factors to the widespread variation and inequities observed in individual wellbeing [3,4]. Latino/a-origin populations in the United States are documented to face a disproportionate burden of both incidence of and complications from many diseases, including diabetes and liver disease [31,32]. From the lens of social determinants of health, they face some of the largest disparities among U.S.-based ethnic groups in areas including inequities in health care access, education, and occupational protections [33]. However, Latino/as also exhibit lower overall mortality rates as compared to non-Hispanic Whites, a phenomenon known as the “immigrant epidemiological paradox” [34]. The potential for place-based factors to act as sources of both added stress and mitigating resilience among U.S.-based Latino/as is a topic that warrants further exploration [1,35,36].

In this article we explore community perceptions of Mexican origin residents who live in rural Yuma County, Arizona. This locale merits study due to both the under-reporting of border residents’ (and especially foreign-born residents’) community perceptions and because of the common media portrayals of this region as plagued by violence and disorder [13]. The fact that a majority of residents of this region are foreign born and have regular cross-border contact facilitates a binational study framework that brings to the forefront the important role of external referents as influencers of individuals’ community perceptions and health promoting practices [3].

Findings reported here reveal that participants overwhelmingly perceived their rural communities in southern Yuma County, Arizona as both calm and safe. While these results contradict popular media characterizations of the borderlands, they are consistent with crime statistics demonstrating an inverse relationship between crime and both border proximity and high immigrant density [37,38]. In addition, our results mirror findings from Castañeda and Chiappetta’s (2020) quantitative study in the border city of El Paso, Texas, in which 42.3% of survey participants used the word “calm” to describe the area and 97% reported feeling safe or very safe [13].

Study participants’ descriptions of calm were primarily tied to their homes being in small towns that had a slow pace of life, were walkable for basic needs, and promoted a low stress environment that encouraged participation and ease in their communities. These findings reflect prior research showing that walkability enhances opportunities for recreation and community engagement [10]. Participants also linked feelings of calm to easeful relationships with their primarily Mexican origin neighbors and familiarity with the region. The multiple benefits of the “ethnic enclave effect” have been previously documented, including increased social control and decreased all-cause mortality and morbidity related to stroke, cancer, and hypertension [18,39]. The fact that participants’ communities in Yuma County boasted both a largely Mexican population and proximity to Mexico meant that they provided a positive balance of physical safety and cultural familiarity. This resulted in participants being able to appreciate perceived safety and calm in Yuma County while also taking advantage of the opportunities for sustained cultural and familial ties and maintenance of cultural practices.

Participants’ perceptions of calm in rural Yuma County were partially formed in relationship to external referents developed in large metropolises in both Mexico and U.S. border states where they had previously worked or lived. Study staff from Campesinos Sin Fronteras noted with interest participants’ common perception of regional calm given that San Luis is second busiest port of entry in the state of Arizona and the primary gateway for Mexican agricultural workers heading north, meaning that significant cross border traffic is a constant presence.

The salience of external referents was even more noticeable in the realm of participants’ perceived safety. Despite substantial and positive cross border ties to family, culture, and hometowns, participants made repeated references to murders, extortion, kidnappings, and drug trafficking in Mexico. Over the past several decades, drug-related violence has escalated in many Mexican communities, fueled by U.S. demand for illegal substances, official crackdown by the Mexican government, and U.S. exports of firearms and military training [40]. This phenomenon has unfolded with particular brutality along Mexico’s northern border, and Mexicali, Baja California Norte and San Luis Río Colorado, Sonora have experienced a dramatic uptick in drug related homicides in recent years [16,17]. Study staff from Campesinos Sin Fronteras reinforced these findings, noting that some residents of San Luis and Somerton, Arizona had been victims of the wave of kidnappings and murders across the border, plunging several families into unspeakable pain and anxiety.

Thus, despite mentioning important safety concerns in Yuma County including drug trafficking and potential spillover violence from Mexico, participants overwhelmingly perceived the area as safe, based on the absence of extreme acts of violence and threats to material belongings. Castañeda and Chiappetta (2020) likewise found that people born outside of El Paso (many of whom were from Mexico) were more likely to describe the El Paso area as safe, concluding that: “This may be because many of those not raised in El Paso are immigrants, and their referents for safety are more unsafe than El Paso” [13] (p. 9).

Luna-Arocas and Tang (2015) argue that the significance of external referents is not uniform but rather enhanced by cultural and other factors that determine the weight assigned to the object of comparison [41]. The regularity with which participants mentioned safety and their willingness to forego the potential advantages of other places in exchange for regional safety indicate that safety and security are of prime importance to Mexican origin border residents. As such, we argue that the comparison between safety levels in the United States and Mexico acted as a central determinant to individuals’ perceptions of their communities in Yuma County. This finding parallels the 1996 World Health Organization’s declaration of violence as a growing global public health crisis that undermines individual and community level health and wellbeing [12,42]. Violence and fear of violence have been linked to decreased physical activity in the neighborhood setting and declines in overall physical and mental well-being [11,43,44].

Lastly, our findings indicated that participants credited both neighbors and local police forces as key collaborators in the promotion of safety in southern Yuma County. Many participants indicated that their neighbors—who were overwhelmingly of Mexican origin—promoted safety by avoiding conflicts among each other and safeguarding each other’s belongings. The social control mechanisms sometimes present in ethnic enclaves have been shown to promote mutual trust and reliance and carry positive implications for health and wellness [13,39]. More specifically, a sense of “collective efficacy”—shared norms around taking action to ensure community wellness—has been linked to lower actual crime rates and perceptions of crime, as well as more distally to decreases in obesity and enhanced opportunities for physical activity [11,45,46].

The fact that the Mexican origin participants of this study widely heralded the active police presence and vigilance in their communities warrants close examination. Mexican immigrants’ negative interactions with police and immigration patrol in the form of daily encounters and immigration raids have been repeatedly shown to generate widespread fear and carry negative health implications [47,48]. Moreover, in a representative sample of residents surveyed in this same region over a decade prior, members of this study team found that residents frequently complained of the overbearing presence of law enforcement in the region, a presence that was characterized by structural racism [30].

Several factors may help explain positive police perceptions in this sample. First, the fact that most study participants had legal residency status in the United States (as evidenced by regular border crossings at official ports of entry) meant they had the ability to engage with police forces without significant fear of deportation or immigration-related detainment. This may explain the contrast between our results and other studies which found that immigrants are frequent targets of theft and other crimes and that they report those crimes less frequently to the police [21,49]. Our respondents were also well-established in their communities and the majority had long-term employment.

Study partners from CSF noted several possible explanations for this finding. They commented that few undocumented people remain in rural Yuma County given the regular border surveillance. In addition, they noted that participants’ lack of complaints related to immigration enforcement authorities may be related to the closures of check-points in the region and a decline in community-level immigration vigilance due to the increasing impermeability of the international border wall as well as a shift in focus toward the surge of Central American refugees. Lastly, they posited that residents’ trust in local police agencies may have been boosted by the fact that most members of the police forces in the region were of Mexican descent and lived locally in the communities they police. Community partners identified many local outreach and support efforts undertaken by the local police forces and identified by name many police officers who were highly appreciated for their community-based efforts. As evidenced by prior work on community policing, improved trust in local police forces is an important predictor of their perceived efficacy [50].

And finally, the role of external referents was again highly relevant in determining participants’ positive perceptions of the local police agencies and the credit they ascribed to them for keeping the region safe and orderly. Community partners described the role of binational referents as a mirror effect, in which the negative reference point of the police forces in San Luis Rio Colorado and other border regions in Sonora improved the way participants viewed U.S. based police forces. They maintain that police forces in San Luis Río Colorado are known as corrupt for accepting bribes and that they are ineffective in fulfilling the duties of public security, sending a message to the public that nothing can stop or punish criminals who act with impunity in broad daylight. Corruption of Mexican armed forces more broadly has been previously shown to undermine popular trust and support [51]. Due to this juxtaposition, the regular surveillance, professionalism, and responsiveness of police forces in Yuma County provided a welcome contrast to experiences pre-migration and/or during cross border visits.

### Study Limitations

There are limitations to this study. Our sample was dominated by people with legal residency status in the United States and thus does not adequately reflect the experiences of undocumented people who would likely feel significantly less safe in this region especially in regards to police and border patrol surveillance. Additionally, we conducted the interviews during a wave of COVID-19 infections that hit the border area disproportionately, reducing opportunities for contact with extended networks in the community. The experiences related to the pandemic thus tended to dominate our conversations regarding sources of stress and resilience.

## 5. Conclusions

Given demonstrated inequities in morbidity and mortality amongst U.S.-based Latino/as, it is imperative to identify environmental factors that may exacerbate health stressors or alternately promote resilience, wellbeing, and positive health practices among Latino/a sub-groups. This article has demonstrated the potential for perceived safety and calm to serve as important contributors to health supporting environments among Mexican origin people living in the U.S. borderlands. Furthermore, threats of violence in Mexico may act to accentuate perceived safety among people of Mexican origin in the U.S. borderlands, indicating an important role for external—and specifically binational—referents in driving community perceptions in this region. However, the undermining of public safety in northern Mexico has the potential to erode the benefits of safe and calm borderland environments by minimizing immigrants’ ability to sustain contacts with family and culture that serve to reduce social isolation and promote cultural continuity. In addition, Mexican immigrants’ opportunities to benefit from secure local environments in the U.S. borderlands are highly contingent on their legal status and their ability to become integrated members of their communities with the full range of civil rights.

## Figures and Tables

**Table 1 ijerph-19-08399-t001:** Characteristics of Mexican Origin Study Participants.

Characteristics	Median/n	%/Range
Age (years)	35	21, 60
Females (% yes)	17	56.6
Married (% yes)	15	50.0
Have Children (% yes)	23	76.6
**Place of Origin**		
From Mexican Border States	23	76.6
From other Mexican States	3	10.0
From U.S. Border States	4	13.3
**Place of Current Residence**		
San Luis, Arizona	17	56.6
Somerton, Arizona	6	20.0
Yuma, Arizona	3	10.0
San Luis Río Colorado, Mexico	4	13.3
**Employment**		
Employed (% yes)	25	83.3
Teaching or Aide (% yes)	5	16.6
Agriculture (% yes)	8	26.6

## Data Availability

Not applicable.

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
