# Peer review of "“Es Muy Tranquilo Aquí”: Perceptions of Safety and Calm among Binationally Mobile Mexican Immigrants in a Rural Border Community"

_ijerph, 2022, doi:10.3390/ijerph19148399_

Round 1
Reviewer 1 Report
Dear authors,
first of all, congratulations for this important study. Studying vulnerable populations is so important and socially relevant!
I have some suggestions to improve even more your manuscript:
1) I the abstract, you should mention the aims of your study before describing the methods.
2) In line 94 when you say"In this article, we aim to redress that short
coming by presenting qualitative data on perceptions of Mexican-origin border residents in southern Yuma County, Arizona", maybe you could say: this manuscript presents the results of a study that aimed to identify the perceptions of Mexican-origin border residents about........."
3) Delete the point 2.1. in line 98, because you should as I suggested incorporate the aims (better explained) in the end of the introduction.
4) In line 115 you refer "Development of the qualitative aim of this study began in August 2020 with weekly meetings" This was a focus group alike methodology? Did you made a script for this meetings? If so, how it was structured?
5) In line 125 you explain that "Based on themes identified in team discussionss, the lead author, developed the qualitative interview guide in English and Spanish. How was this process? the last method was an apriori categorization process to identify the dimensions of the interview guide? I know you can make this more clear.
5) Make sure you make an english review... For e.g. (line 149- "CSF promotoras" should be "CSF promoters". I understand that promotoras was the original name, but in the manuscript is better if you translate.
6) In the supplementary documents you have the original Quotes in Spanish. I was expecting to find the categorization process to the results description and before to the interview guidance explained. Maybe you can add the interview script to suplementary documents and also the categorization definition for the results. It would give a better understanding to readers.
If you consider this suggestions your study will be perfectally valued. Don't give up, a few more steps to perfection!
Author Response
Dear reviewer,
Thank you for your careful read of this manuscript and your helpful suggestions. We respond to each of your points below:
- We added a sentence on the qualitative study aims of the study to the abstract
- We adopted your suggested wording for the final sentence of the introduction
- We have elected to keep 2.1 in the methods because it applies to the larger mixed methods project of which this first qualitative aim is just one phase.
- We have added a sentence explaining that these conversations were guided by the stated study aims but were structured as an open conversation among the study team.
- We have elected not to change "promotoras" to "promoters" as the Spanish name is widely used in both community and academic parlance.
- Thank you for this suggestion and we have added both the bilingual interview guide and the code tree to the appendices.
Reviewer 2 Report
I like this paper. It contributes new data that will be appreciated by many in the scholarly and broader communities.
1. The paper explores the question of whether residents, especially residents of Mexican descent, feel safe and personally supported in communities close to the Mexican border. There is a great deal of public commentary about these border areas being unsafe and the paper explores whether this is perceived by residents. Contrary to broad belief, residents feel safe and secure in these communities.
2. I belief the authors who say that the question of whether border residents feel safe has received little research. The authors place this discussion in the context of health discussions about how community safety and feelings of communalism affect health measures like blood pressure. The authors come at this research from a public health perspective and I think this is appropriate. I also think they carry out the research with competence and thoroughness.
3. There is relatively little research on how community context affects the health of recent Mexican-American migrants. There is particularly little research on the way community is experienced by members of Mexican ethnic enclaves close to the Mexican border. Adding research data on this topic is an important addition to community research.
4. The methodology was carefully designed, described, and executed. I have not criticisms.
5. The data collection and findings are consistent with the theoretical development given in the first part of the paper. Overall, the paper is well written, well argued, and well constructed.
6. The paper includes a thorough and appropriate review of literature.
7. This is a qualitative paper so the only table they have gives descriptive statistics and this is clearly and appropriately presented.
My overall feeling is that the data developed and presented in this paper is pretty simple and descriptive and someone might ask for a more theorized presentation. However, I think that in the first part of the paper where other research is reviewed and theoretical concepts are developed, that the authors make a good case for the relatively straightforward design of data collection and presentation that they use. The authors conveyed the feeling that they were highly competent researchers, that they had thorough knowledge of the field, and that this incremental research project was appropriate to their goals. I'd be bothered by reviewers who wanted them to make more of what they have in terms of producing a more complex and elaborate paper.
Author Response
Dear Reviewer,
Thank you for your careful read of this article and your helpful insights about its contribution. We appreciate your time and your positive feedback.
Round 2
Reviewer 1 Report
Dear authors, thank you for your anwers, clarification and for the improvements you have made in your manuscript.
Just one more suggestion. You don't need to say "qualitative aim", because the qualitative or quantitative are the methods that are used to answer the aims. :) Aims are aims. Do you agree with me?
As "promotoras" are, as you say, the terminology used, maybe you can explain what it means to the readers that are not so familiarized with spanish. But it is an option of your team! :)
Thank you!